# β-Carotene accelerates the resolution of atherosclerosis in mice

**Ivan Pinos[1†], Johana Coronel[2†], Asma'a Albakri[1,3†], Amparo Blanco[1], Patrick McQueen[1], Donald Molina[2], JaeYoung Sim[2], Edward A Fisher[4], Jaume Amengual[1,2]***

[1]Division of Nutritional Sciences, University of Illinois Urbana Champaign, Urbana, United States; [2]Department of Food Science and Human Nutrition, University of Illinois Urbana Champaign, Urbana, United States; [3]The University of Jordan, School of Agriculture, Department of Nutrition and Food Technology, Amman, Jordan; [4]The Leon H. Charney Division of Cardiology, Department of Medicine, The Marc and Ruti Bell Program in Vascular Biology, New York University Grossman School of Medicine, NYU Langone Medical Center, New York, United States

**\*For correspondence:**
jaume6@illinois.edu

†These authors contributed equally to this work

**Competing interest:** The authors declare that no competing interests exist.

**Abstract** β-Carotene oxygenase 1 (BCO1) catalyzes the cleavage of β-carotene to form vitamin A. Besides its role in vision, vitamin A regulates the expression of genes involved in lipid metabolism and immune cell differentiation. BCO1 activity is associated with the reduction of plasma cholesterol in humans and mice, while dietary β-carotene reduces hepatic lipid secretion and delays atherosclerosis progression in various experimental models. Here we show that β-carotene also accelerates atherosclerosis resolution in two independent murine models, independently of changes in body weight gain or plasma lipid profile. Experiments in *Bco1⁻/⁻* mice implicate vitamin A production in the effects of β-carotene on atherosclerosis resolution. To explore the direct implication of dietary β-carotene on regulatory T cells (Tregs) differentiation, we utilized anti-CD25 monoclonal antibody infusions. Our data show that β-carotene favors Treg expansion in the plaque, and that the partial inhibition of Tregs mitigates the effect of β-carotene on atherosclerosis resolution. Our data highlight the potential of β-carotene and BCO1 activity in the resolution of atherosclerotic cardiovascular disease.

## eLife assessment

This study presents an **important** conceptual advance of how vitamin A and its derivatives contribute to atherosclerosis. There is **solid** evidence for the contributions of specialized populations of T cells in atherosclerosis resolution, including use of multiple in vivo models to validate the functional effects. A limitation is the insufficient analysis of lesions, but the manuscript has been improved from the original preprint version and the overarching conclusions have been refined.

## Introduction

Atherosclerotic cardiovascular disease is a progressive pathological process initiated by the accumulation of cholesterol-rich lipoproteins within the intima layer of the arterial wall. These particles ultimately lead to the production of chemoattractant cues for circulating monocytes that transmigrate across the endothelial layer to reach the intima and then differentiate to macrophages to become cholesterol-laden foam cells. During lesion development, macrophages promote local inflammation and plaque weakening by degrading extracellular matrix components such as collagen fibers, which can evolve in the rupture of the lesion and thrombus formation (*Newby et al., 2009*).

Conventional therapies aim to lower plasma cholesterol to mitigate the progression of atherosclerosis. Novel strategies are currently under development to stimulate the resolution of plaque inflammation more directly, and eventually a reduction in lesion size, in a process named atherosclerosis regression (*Goldberg et al., 2020*). Among these strategies, the modulation of CD4[+] regulatory T cells (Tregs) number is gaining interest over the past years since the discovery that regressing lesions are enriched in Tregs, where they promote plaque stabilization and repair (*Andersson et al., 2010*; *Witztum and Lichtman, 2014*). Tregs typically express the surface marker CD25, which has been utilized to target and eliminate Tregs (*Onda et al., 2019*; *Sharma et al., 2020*). However, strategies to deplete CD25[+] Tregs do not affect CD25[-] Tregs, which possess similar immunomodulatory properties as CD25[+] Tregs (*Couper et al., 2007*). Among the different Tregs markers, the forkhead box P3 (FoxP3) acts as lineage specification factor regulating gene expression of proteins implicated in the immunosuppressive activity of these cells (*Fontenot et al., 2005*). Cell culture studies show that retinoic acid, the transcriptionally active form of vitamin A, promotes Treg differentiation by upregulating FoxP3 expression (*Elias et al., 2008*); however, whether dietary vitamin A affects Tregs during atherosclerosis development and resolution remains unanswered.

Humans obtain vitamin A primarily from β-carotene and other provitamin A carotenoids present in most fruits and vegetables. Upon absorption, provitamin A carotenoids are cleaved by the action of β-carotene oxygenase 1 (BCO1), the limiting enzyme in vitamin A formation (*von Lintig and Vogt, 2000*). Our data show that the enzymatic activity of BCO1 mediates the bioactive actions of β-carotene in various preclinical models of obesity and atherosclerosis (*Hessel et al., 2007*; *Amengual et al., 2013*; *Amengual et al., 2011*; *Lobo et al., 2010*; *Zhou et al., 2020*; *Coronel et al., 2022*). We showed that the dietary supplementation with β-carotene delays atherosclerosis progression by reducing cholesterol hepatic secretion in low-density lipoprotein receptor (LDLR)-deficient (*Ldlr*[-/-]) mice (*Zhou et al., 2020*). We also reported that subjects harboring a genetic variant linked to greater BCO1 activity were associated with a reduction in plasma cholesterol (*Amengual et al., 2020*).

In this study, we tested the effect of dietary β-carotene on atherosclerosis resolution in two independent experimental models. We characterized plaque composition by probing for macrophage and collagen contents, two parameters utilized to characterize atherosclerotic lesions undergoing resolution (*Amengual et al., 2021*; *Josefs et al., 2021*; *Josefs et al., 2020*; *Barrett et al., 2019*). Lastly, we utilized *Bco1*[-/-] mice and CD25[+] Treg depletion experiments in *Foxp3*[EGFP] mice to tease out the direct implications of vitamin A formation and Tregs on atherosclerosis resolution.

## Results

### β-Carotene supplementation accelerates atherosclerosis resolution

We recently showed that dietary β-carotene delays atherosclerosis progression in *Ldlr*[-/-] mice (*Zhou et al., 2020*), which prompted us to examine whether β-carotene also impacts the resolution of inflammation in complex atherosclerotic lesions. To achieve these lesions, we utilized two distinct mouse models of atherosclerosis in combination with WD-VAD. In our first model, we established LDLR deficiency in wild-type mice by injecting anti-sense oligonucleotide targeting the low-density lipoprotein receptor (ASO-LDLR) weekly for a period of 16 wk. The characteristics of the plaques before undergoing resolution were established by sacrificing a subset of mice after 16 wk on diet (baseline). The remaining mice were injected once with SO-LDLR to promote atherosclerosis resolution and divided into two groups fed either WD-VAD (resolution–control) or WD-β-carotene (resolution–β-carotene). Mice were harvested 3 wk after SO-LDLR injections (*Figure 1A*).

Male mice gained more weight than female mice, although we did not observe differences between the three experimental groups (*Figure 1—figure supplement 1A*). For the remaining outcomes (plasma lipid profile, retinoid levels, and plaque composition), we did not observe sex differences in any of our experiments, and therefore, we combined results from both sexes. Both resolution groups, independently of β-carotene, showed drastic reduction in total plasma cholesterol and triglyceride levels in comparison to the baseline group. HDL-C levels remained constant between groups (*Figure 1B*). Using pooled samples, we compared plasma cholesterol and triglyceride partitioning between groups by fast performance liquid chromatography (FPLC). Very low-density lipoprotein (VLDL) and LDL fractions in both resolution groups showed a comparable reduction in total

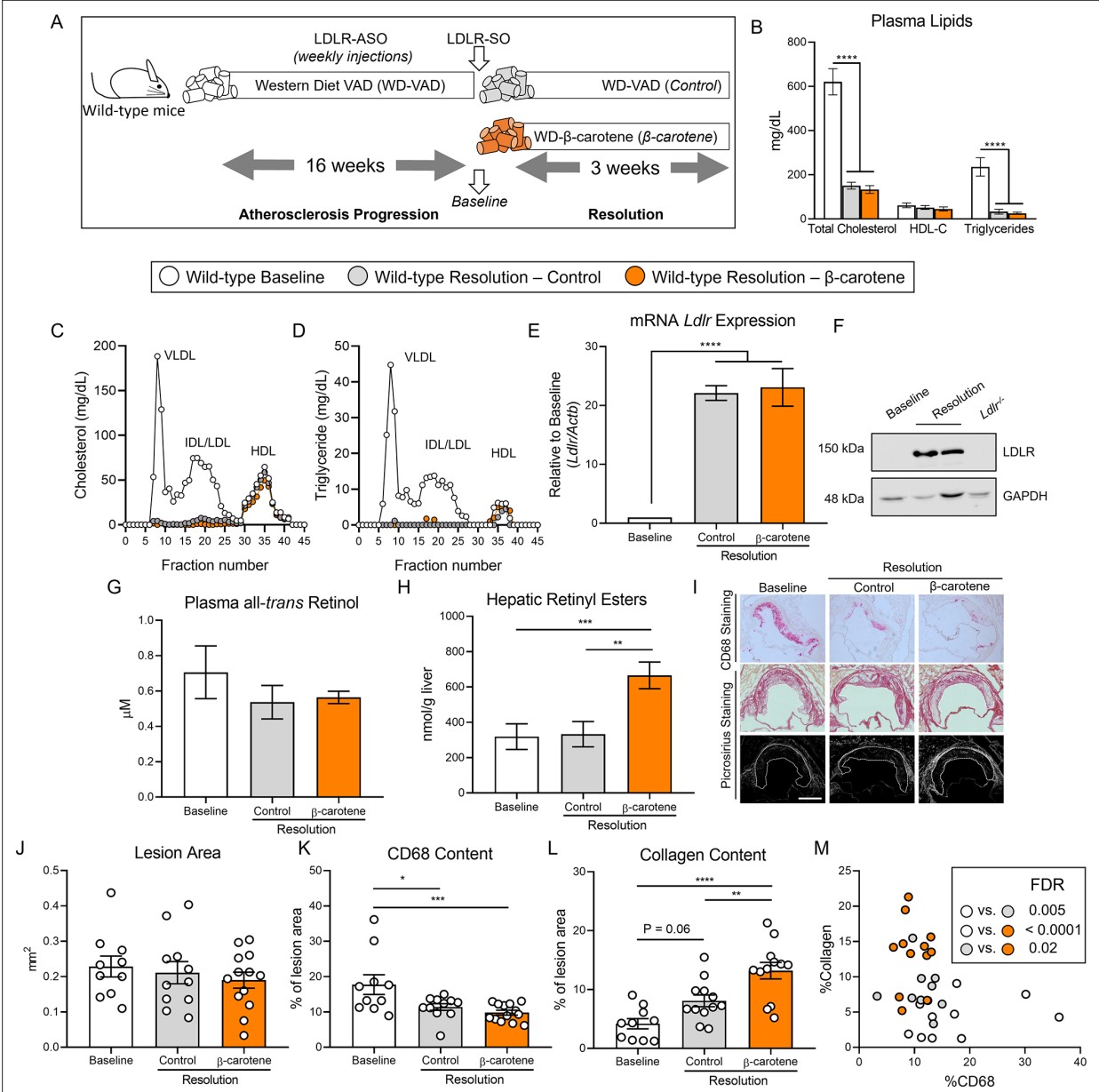

**Figure 1.** β-Carotene accelerates atherosclerosis resolution in wild-type mice infused with antisense oligonucleotide targeting the low-density lipoprotein receptor (ASO-LDLR). (**A**) Four-week-old male and female wild-type mice were fed a purified Western diet deficient in vitamin A (WD-VAD) and injected with ASO-LDLR once a week for 16 wk to induce atherosclerosis. After 16 wk, a group of mice was harvested (baseline) and the rest of the mice were injected once with sense oligonucleotide (SO-LDLR) to inactivate ASO-LDLR and promote atherosclerosis resolution. Mice undergoing resolution were either kept on the same diet (resolution–control) or switched to a Western diet supplemented with 50 mg/kg of β-carotene (resolution–β-carotene) for three more weeks. (**B**) Plasma lipid levels at the moment of the sacrifice. (**C**) Cholesterol and (**D**) triglyceride distribution in fast performance liquid chromatography (FPLC)-fractioned plasma (data pooled from five mice/group). (**E**) Relative low-density lipoprotein receptor (LDLR) mRNA, and (**F**) protein expressions in the liver. (**G**) Circulating vitamin A (all-*trans* retinol), and (**H**) hepatic retinyl ester stores determined by HPLC. (**I**) Representative images for macrophage (CD68+, top panels) and picrosirius staining to identify collagen using the bright-field (middle panels) or polarized light (bottom panels). (**J**) Plaque size, (**K**) relative CD68 content, and (**L**) collagen content in the lesion. (**M**) Descriptive discriminant analysis employing the relative CD68 and collagen contents in the lesion as variables highlighting the false discovery rate (FDR) for each comparison. Each dot in the plot represents an individual mouse (n = 10–12/group). Values are represented as means ± SEM. Statistical differences were evaluated using one-way ANOVA with Tukey's multiple comparisons test. Differences between groups were considered significant with a p-value<0.05. *p<0.05; **p<0.01; ***p<0.005; ****p<0.001. Size bar = 200 µm.

The online version of this article includes the following figure supplement(s) for figure 1:

**Figure supplement 1.** Four-week-old male and female wild-type mice were fed a purified Western diet deficient in vitamin A (WD-VAD) and injected with antisense oligonucleotide targeting the low-density lipoprotein receptor (ASO-LDLR) once a week for 16 wk to induce atherosclerosis.

cholesterol and triglyceride content in comparison to baseline mice (*Figure 1C and D*). Lipid normalization was accompanied by the upregulation of hepatic LDLR expression at the mRNA and protein levels (*Figure 1E and F*).

Despite mice being fed a WD-VAD for several weeks, HPLC measurements of circulating vitamin A and hepatic retinoid stores ruled out vitamin A deficiency in any of the experimental groups (*Figure 1G and H*). However, mice fed WD-β-carotene showed an increase in hepatic vitamin A content in comparison to those fed WD-VAD, which was mediated by the conversion of β-carotene to vitamin A (*Figure 1H*). Hepatic *Cyp26a1* expression, which is commonly utilized as a surrogate marker of vitamin A and retinoic acid production (*Emoto et al., 2005*; *Abu-Abed et al., 2002*), appeared upregulated in response to WD-β-carotene (*Figure 1—figure supplement 1B*).

To determine the effect of β-carotene supplementation on atherosclerosis resolution, we examined lesion size and composition at the level of the aortic root (*Figure 1I*). Lesion area remained constant among the three experimental groups as previously reported in mice treated with ASO-LDLR under similar experimental conditions (*Figure 1J*; *Basu et al., 2018*). We next examined plaque composition by focusing on two parameters commonly utilized as a surrogate indicators of atherosclerosis resolution: CD68$^+$ area, a myeloid (macrophage) marker that represents plaque inflammation, and collagen area, a marker of plaque stability in humans (*Libby and Aikawa, 2002*; *Soehnlein and Libby, 2021*). In comparison to baseline mice, the lesions of resolution–control and resolution–β-carotene groups presented a reduction in CD68 content of approximately 36 and 45%, respectively (*Figure 1K*). The collagen content in lesions increased in both resolution groups in comparison to baseline, reaching statistical significance only in the β-carotene-fed mice. In this group, we observed 215 and 60% increase in collagen content compared to the baseline and the resolution–control groups, respectively (*Figure 1L*). Lastly, we plotted the relative CD68 and collagen contents in the lesion to perform a descriptive discriminant analysis. Individual samples from three distinct experimental groups clustered together, highlighting significant differences between the three experimental groups for all the comparisons (*Figure 1M*).

To validate the results obtained in our ASO-LDLR reversible model of atherosclerosis, we utilized *Ldlr*$^{-/-}$ mice fed WD-VAD subjected to a dietary switch strategy to lower cholesterol in plasma and promote atherosclerosis resolution (*Amengual et al., 2021*; *Yu et al., 2017*). Baseline *Ldlr*$^{-/-}$ mice were harvested after 12 wk on WD-VAD, while the remaining animals were switched to either a standard diet without vitamin A (resolution–control) or the same diet supplemented with β-carotene (resolution–β-carotene) for four more weeks. To prevent changes in food intake due to consistency and hardness of the feed, we provided standard diets as powder. This approach prevented a reduction in body weight, which could have resulted in confounding alterations in atherosclerosis resolution (*Figure 2—figure supplement 1A*).

*Ldlr*$^{-/-}$ mice in both resolution groups presented a comparable reduction in plasma cholesterol and triglyceride levels in comparison to the baseline group. These changes were not accompanied by alterations in HDL-C levels between groups (*Figure 2A*). Systemic and hepatic vitamin A levels failed to show indications of vitamin A deficiency, and hepatic vitamin A stores increased in β-carotene-fed mice (*Figure 2—figure supplement 1B*).

We next characterized atherosclerotic lesions at the level of the aortic root (*Figure 2B*). Lesion size area was comparable between experimental groups (*Figure 2C*), although CD68 content decreased to the same extent in both resolution groups in comparison to baseline mice (*Figure 2D*). In comparison to the baseline group, collagen accumulation in the lesion increased 100% in the resolution–control and over 200% in the resolution–β-carotene group, respectively (*Figure 2E*). Descriptive discriminant analysis showed comparable results to those observed for our ASO-LDLR model, where differences between the three experimental groups reached statistical significance (*Figure 2F*).

Together, these data show that β-carotene supplementation during atherosclerosis resolution results in the improvement of lesion composition in two independent mouse models.

## BCO1 drives the effect of β-carotene on atherosclerosis resolution

To establish the contribution of BCO1 on the effect of β-carotene on atherosclerosis resolution, we utilized LDLR-ASO to promote atherogenesis in *Bco1*$^{-/-}$ mice following the same experimental approach described for wild-type mice (*Figure 1A*). Consistent with our results in our two resolution models (*Figures 1B and 2A*), plasma cholesterol levels decreased in both resolution groups in

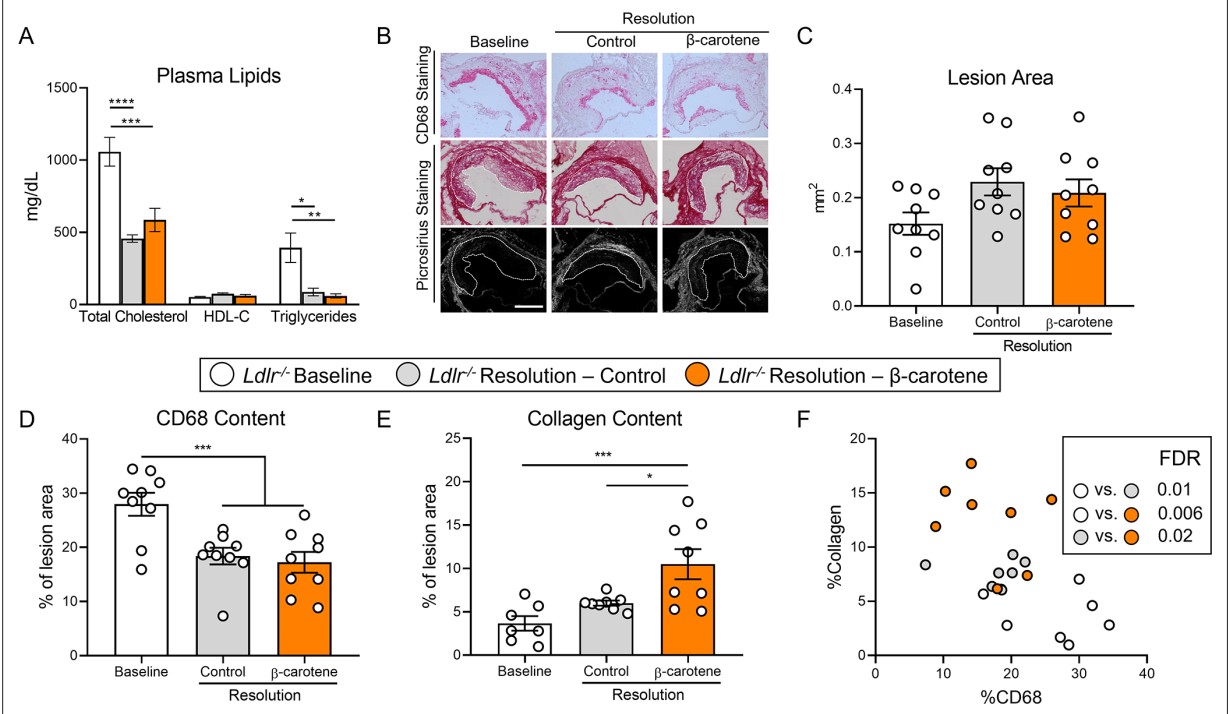

**Figure 2.** β-Carotene accelerates atherosclerosis resolution in low-density lipoprotein-deficient (*Ldlr⁻/⁻*) mice subjected to dietary switch. Four-week-old male and female *Ldlr⁻/⁻* mice were fed a purified Western diet deficient in vitamin A (WD-VAD) for 12 wk to induce atherosclerosis. After 12 wk, a group of mice was harvested (baseline) and the rest of the mice were switched to a standard diet (resolution–control) or the same diet supplemented with 50 mg/kg of β-carotene (resolution–β-carotene) for four more weeks. (**A**) Total cholesterol plasma levels at the moment of the sacrifice. (**B**) Representative images for macrophage (CD68⁺, top panels) and picrosirius staining to identify collagen using the bright-field (middle panels) or polarized light (bottom panels). (**C**) Plaque size, (**D**) relative CD68 content, and (**E**) collagen content in the lesion. (**F**) Descriptive discriminant analysis employing the relative CD68 and collagen contents in the lesion as variables highlighting the false discovery rate (FDR) for each comparison. Each dot in the plot represents an individual mouse (n = 9–12/group). (**A–E**) Values are represented as means ± SEM. Statistical differences were evaluated using one-way ANOVA with Tukey's multiple comparisons test. Differences between groups were considered significant with a p-value<0.05. *p<0.05; ***p<0.005; ****p<0.001. Size bar = 200 μm.

The online version of this article includes the following figure supplement(s) for figure 2:

**Figure supplement 1.** Four-week-old male and female *Ldlr⁻/⁻* mice were fed a purified Western diet deficient in vitamin A (WD-VAD) for 12 wk to induce atherosclerosis.

comparison to baseline mice (***Figure 3—figure supplement 1A***). Similarly, HPLC measurements of plasma and hepatic vitamin A ruled out vitamin A deficiency in *Bco1⁻/⁻* mice baseline and resolution mice (***Figure 3—figure supplement 1B and C***). The accumulation of β-carotene in tissues and plasma is characteristic in *Bco1⁻/⁻* mice (***Hessel et al., 2007***). Indeed, HPLC quantification of β-carotene in plasma and liver of *Bco1⁻/⁻* resolution– β-carotene mice presented 4-fold and 400-fold greater β-carotene levels found in wild-type resolution–β-carotene, respectively (***Figure 3A and B***).

The characterization of the atherosclerotic lesions showed no alterations in lesion size between the three experimental groups (***Figure 3C and D***). When compared to baseline mice, both resolution groups displayed lower CD68 and higher collagen contents, although we did not observe differences in CD68 and collagen contents between both resolution groups (***Figure 3E and F***). Descriptive discriminant analysis highlighted differences between baseline *Bco1⁻/⁻* mice and their resolution littermates, independently of the presence of β-carotene in the diet (***Figure 3G***).

## Effect of β-carotene and anti-CD25 depletion on Treg cell number

Our data show that β-carotene favors atherosclerosis resolution in two independent mouse models (***Figures 1M and 2F***), and results in *Bco1⁻/⁻* mice directly implicate vitamin A formation in this process (***Figure 3G***). Retinoic acid is the transcriptionally active form of vitamin A and promotes Treg differentiation by upregulating FoxP3 expression in various experimental models (***Gundra et al., 2017***;

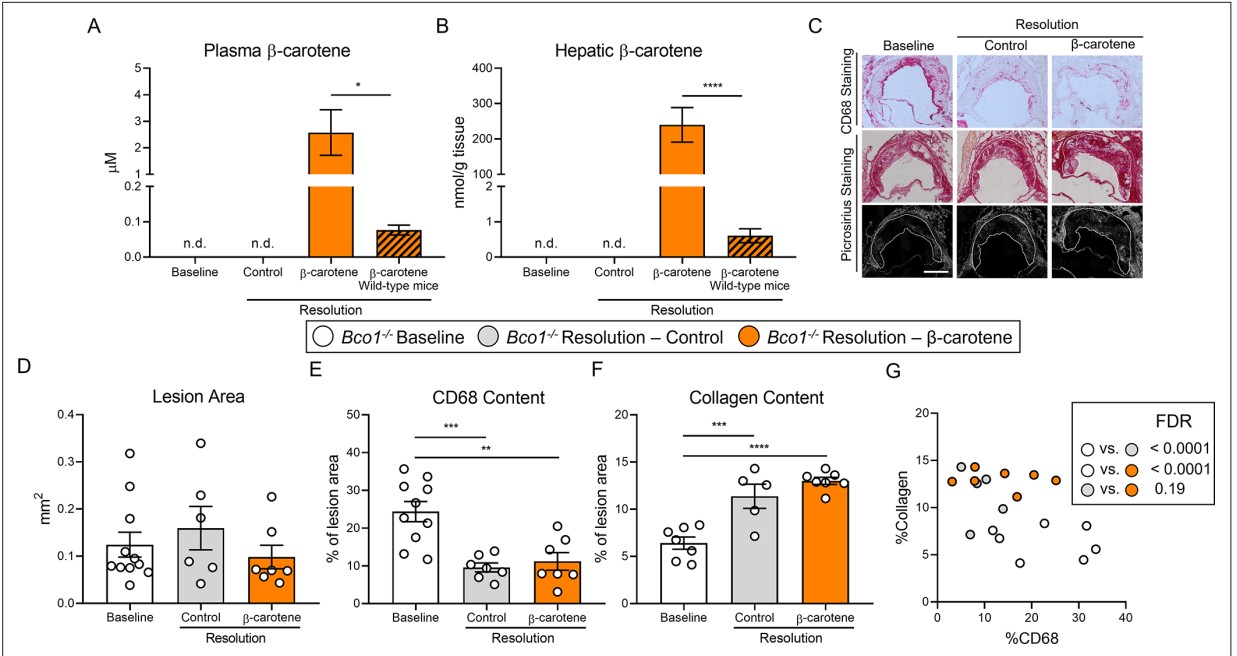

**Figure 3.** β-Carotene supplementation does alter atherosclerosis resolution in β-carotene oxygenase 1-deficient (*Bco1⁻ᐟ⁻*) mice infused with antisense oligonucleotide targeting the low-density lipoprotein receptor (ASO-LDLR). Four-week-old male and female *Bco1⁻ᐟ⁻* mice were fed a purified Western diet deficient in vitamin A (WD-VAD) and injected with ASO-LDLR once a week for 16 wk to induce the development of atherosclerosis. After 16 wk, a group of mice was harvested (baseline) and the rest of the mice were injected once with sense oligonucleotide (SO-LDLR) to block ASO-LDLR (resolution). Mice undergoing resolution were either kept on the same diet (resolution–control) or switched to a Western diet supplemented with 50 mg/kg of β-carotene (resolution–β-carotene) for three more weeks. (**A**) β-Carotene levels in plasma and (**B**) liver at the sacrifice determined by HPLC. (**C**) Representative images for macrophage (CD68⁺, top panels) and picrosirius staining to identify collagen using the bright-field (middle panels) or polarized light (bottom panels). (**D**) Plaque size, (**E**) relative CD68 content, and (**F**) collagen content in the lesion. (**G**) Descriptive discriminant analysis employing the relative CD68 and collagen contents in the lesion as variables highlighting the false discovery rate (FDR) for each comparison. Each dot in the plot represents an individual mouse (n = 5–11/group). (**A–F**) Values are represented as means ± SEM. Statistical differences were evaluated using one-way ANOVA with Tukey's multiple comparisons test. Differences between groups were considered significant with a p-value<0.05. *p<0.05; **p<0.01; ***p<0.005; ****p<0.001. Size bar = 200 µm.

The online version of this article includes the following figure supplement(s) for figure 3:

**Figure supplement 1.** Four-week-old male and female *Bco1⁻ᐟ⁻* mice were fed a purified Western diet deficient in vitamin A (WD-VAD) and injected with antisense oligonucleotide targeting the low-density lipoprotein receptor (ASO-LDLR) once a week for 16 wk to induce the development of atherosclerosis.

*Ouimet et al., 2015*; *Gundra et al., 2014*; *Girgis et al., 2014*; *Girgis et al., 2013*; *Broadhurst et al., 2012*). Treg number decreases in developing lesions and increases during atherosclerosis resolution (*Sharma et al., 2020*; *Foks et al., 2015*; *Ait-Oufella et al., 2006*), but whether the dietary manipulation of β-carotene or vitamin A affects Treg number during atherosclerosis remains unanswered. To investigate whether the Tregs are responsible for the effect of dietary β-carotene on atherosclerosis resolution, we induced atherosclerosis in *Foxp3^EGFP* mice by injecting ASO-LDLR as described above (*Figures 1 and 3*). *Foxp3^EGFP* mice co-express EGFP and FoxP3 under the regulation of the endogenous FoxP3 promoter (*Haribhai et al., 2007*). After 16 wk on diet, mice undergoing resolution were treated twice with either the PC61 anti-CD25 monoclonal antibody (anti-CD25) or an isotype control (IgG) (*Figure 4A*; *Sharma et al., 2020*). We did not observe changes in body weight among groups, and the three groups undergoing resolution showed a normalization in plasma lipids in comparison to baseline controls (data not shown).

Flow cytometry analyses demonstrated the effectiveness of the anti-CD25 treatment by reducing the ratio of circulating and splenic CD25⁺FoxP3⁺ (CD25⁺ Tregs) in comparison to the other experimental groups (*Figure 4B–D*). We also observed a depletion of CD25⁺ Tregs and CD25⁺FoxP3⁻ T cells in the lesion and lymph nodes (*Figure 4E and F*), in agreement with previous reports showing that anti-CD25 fails to completely deplete CD25⁻FoxP3⁺ Tregs (CD25⁻ Tregs), which retain strong anti-inflammatory properties (*Couper et al., 2007*). Hence, we quantified the number of total Tregs

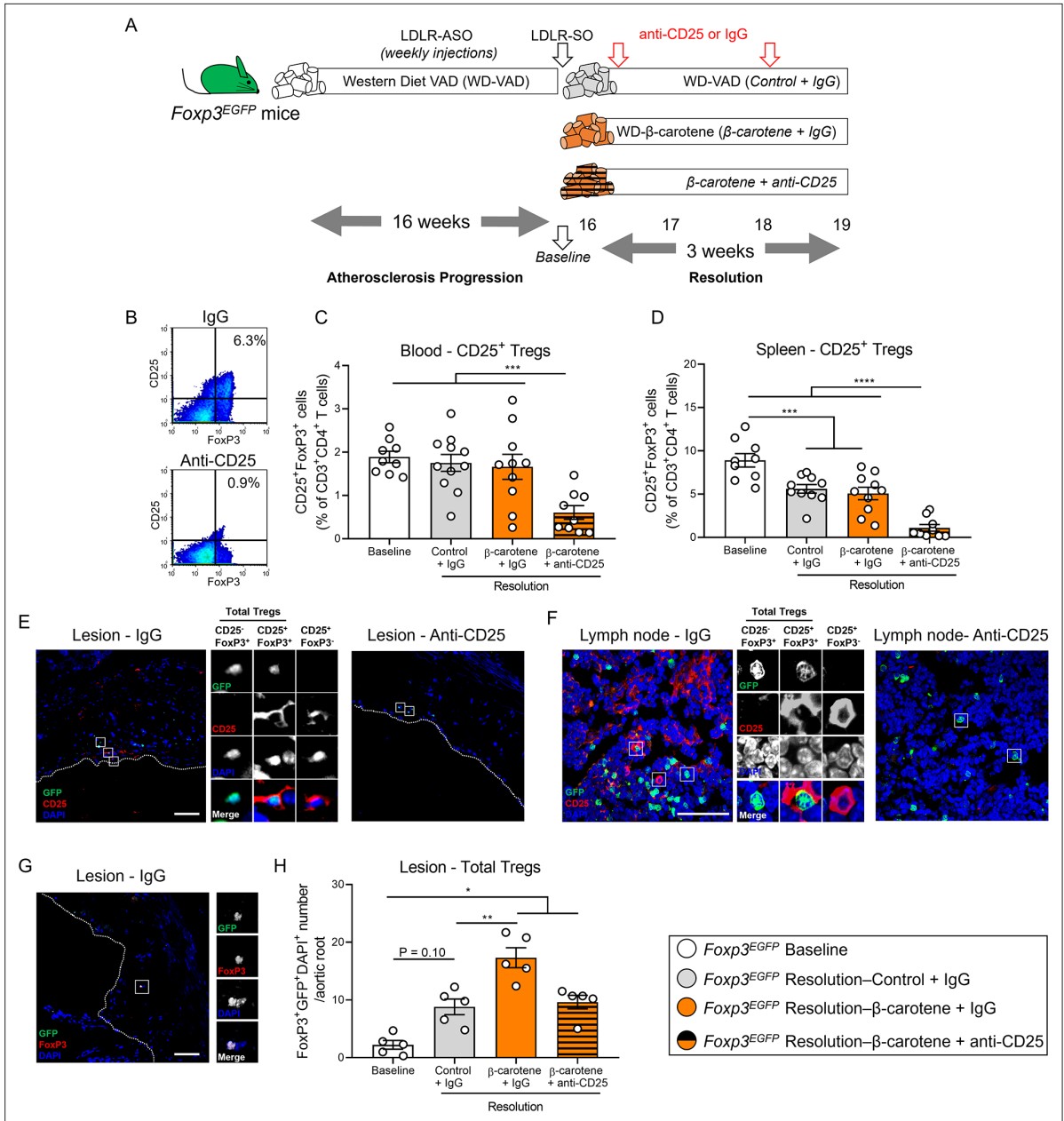

**Figure 4.** Effect of anti-CD25 treatment on Treg number. (**A**) Four-week-old male and female mice expressing enhanced green fluorescence protein (EGFP) under the control of the forkhead box P3 (*Foxp3*) promoter (*Foxp3^EGFP* mice) were fed a purified Western diet deficient in vitamin A (WD-VAD) and injected with antisense oligonucleotide targeting the low-density lipoprotein receptor (ASO-LDLR) once a week for 16 wk to induce the development of atherosclerosis. After 16 wk, a group of mice was harvested (baseline) and the rest of the mice were injected once with sense oligonucleotide (SO-LDLR) to block ASO-LDLR (resolution). Mice undergoing resolution were either kept on the same diet (resolution–control) or switched to a Western diet supplemented with 50 mg/kg of β-carotene (resolution–β-carotene) for three more weeks. An additional group of mice fed with β-carotene was injected twice before sacrifice with anti-CD25 monoclonal antibody to deplete Treg (resolution–β-carotene+anti-CD25). The rest of the resolution groups were injected with IgG isotype control antibody. (**B**) Representative flow cytometry panels showing splenic CD25+FoxP3+ (CD25+ Treg) cells in mice injected with IgG or anti-CD25. (**C**) Quantification of the splenic and (**D**) circulating blood levels of CD25+ Treg cells determined by flow cytometry. (**E**) Representative confocal images show the presence of total Tregs (CD25-FoxP3+ + CD25+ Tregs) and CD25+FoxP3- T cells in the lesion of mice injected with IgG (left panel) or anti-CD25 (right panel). (**F**) Representative confocal images show the presence of total Tregs and CD25+FoxP3- T cells in lymph nodes of mice injected with IgG (left panel) or anti-CD25 (right panel). quantification of CD25+ Tregs in the lesions. (**G**) Representative confocal image and (**H**) quantification of total Tregs in the lesion. Each dot in the plot represents an individual mouse (n = 5–11 mice/group). Values are represented as means ± SEM. Statistical differences were evaluated using one-way ANOVA with Tukey's multiple comparisons test. Differences between groups were considered significant with a p-value<0.05. *p<0.05; **p<0.01; ***p<0.005; ****p<0.001.

*Figure 4 continued on next page*

*Figure 4 continued*

The online version of this article includes the following figure supplement(s) for figure 4:

**Figure supplement 1.** Four-week-old male and female mice expressing enhanced green fluorescence protein (EGFP) under the control of the forkhead box P3 (*Foxp3*) promoter (*Foxp3^EGFP* mice) were fed a purified Western diet deficient in vitamin A (WD-VAD) and injected with antisense oligonucleotide targeting the low-density lipoprotein receptor (ASO-LDLR) once a week for 16 wk to induce the development of atherosclerosis.

independently of the presence of CD25 by counting the number of GFP/FoxP3/DAPI triple-positive cells (*Figure 4G*). All the groups undergoing resolution presented a greater total Treg number than baseline mice, although the results did not reach statistical significance between baseline and resolution–control group (p=0.10). Resolution–β-carotene injected with IgG presented the highest Treg cell number among all the other experimental groups, suggesting that dietary β-carotene favors Treg expansion in the plaque (*Figure 4H*).

Circulating total Treg number remained constant, but we observed a slight decrease in the number of splenic total Tregs in mice undergoing resolution in comparison to baseline mice. This reduction was more pronounced in the resolution–β-carotene mice+anti-CD25 (*Figure 4—figure supplement 1A and B*). The number of circulating CD25$^-$ Tregs increased in the resolution–β-carotene mice+anti-CD25 group in comparison to baseline mice, remaining constant in the spleen among all groups (*Figure 4—figure supplement 1C and D*).

## Anti-CD25 treatment partially abrogates the effect of β-carotene on atherosclerosis resolution

Tregs possess strong anti-inflammatory properties independently of the presence of CD25 (*Couper et al., 2007*; *Hayes et al., 2020*) and play an important role in plaque remodeling during atherosclerosis regression (*Sharma et al., 2020*). Hence, we evaluated plaque composition in our *Foxp3^EGFP* mice (*Figure 5A*). As expected, all experimental groups showed comparable lesion size at the level of the aortic root (*Figure 5B*). Mice in the resolution–β-carotene+IgG group displayed a reduction in CD68 content in the lesion of approximately 50% in comparison to baseline mice. Resolution–control+IgG and resolution–β-carotene+anti-CD25 groups showed a 30% reduction in comparison to baseline mice, although only the former reached statistical significance (*Figure 5C*). Collagen content in the lesion of resolution–β-carotene+IgG mice was significantly higher in comparison to any other experimental group. Resolution control+IgG and resolution–β-carotene+anti-CD25 showed comparable results, while baseline mice had the lowest collagen content among all the groups (*Figure 5D*).

We next performed a descriptive discriminant analysis to examine pairwise comparisons between our four experimental groups. The baseline group was significantly different in comparison to all the resolution counterparts. Resolution–β-carotene+IgG mice were different from their resolution–control+IgG, as we observed in our two previous resolution experiments (*Figures 1M and 2F*). We did not observe statistical differences when we compared resolution–β-carotene+anti-CD25 to resolution–control+IgG (FDR = 0.38) or resolution–β-carotene+IgG to (FDR = 0.10) (*Figure 5E*).

The presence of anti-inflammatory macrophages in the lesions, together with a net egress of proinflammatory macrophages, are key features of atherosclerosis resolution (*Moore et al., 2018*). We quantified arginase 1 content in the lesion, a key anti-inflammatory marker in mice that contributes to collagen synthesis (*Szondi et al., 2021*), and is synergistically upregulated in anti-inflammatory macrophages exposed to retinoic acid (*Yurdagul et al., 2020*; *Pinos et al., 2023*; *Figure 5F*). Only those mice fed β-carotene, independent of the injection with IgG or anti-CD25, showed an upregulation of arginase 1 in the lesion (*Figure 5G*). We next evaluated retinoic acid signaling in the atherosclerotic lesion by quantifying the levels of the retinoic acid-sensitive marker Cyp26b1 in the plaque (*Figure 5H*; *Ross and Zolfaghari, 2011*). We observed an upregulation in Cyp26b1 expression in mice fed β-carotene, independently of receiving IgG or anti-CD25 (*Figure 5I*). We also evaluated macrophage egress by injecting a single dose of EdU a week before sacrificing the baseline mice (see 'Methods' for details; *Figure 5J*). Only those mice fed β-carotene, independent of the antibody treatment, displayed a greater egress in comparison to baseline mice (*Figure 5K*). We did not observe changes in monocyte recruitment evaluated by injecting fluorescently labeled beads 24 hr before the sacrifice (*Figure 5L and M*), nor changes in Ki67$^+$CD68$^+$ cells, as an indicator of macrophage proliferation in the lesion (*Figure 5—figure supplement 1A and B*).

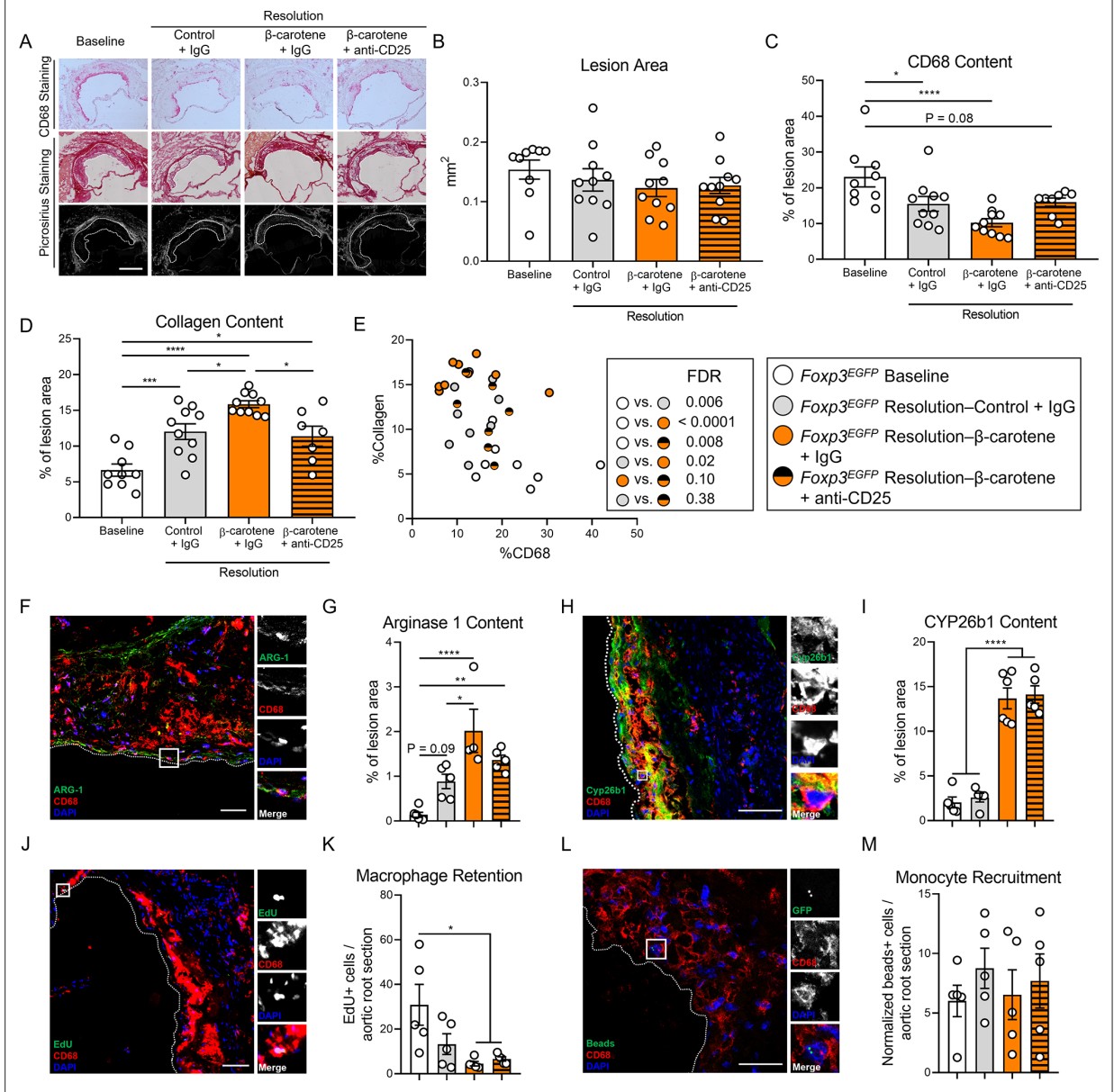

**Figure 5.** Effect of anti-CD25 treatment on lesion composition and monocyte/macrophage trafficking. Four-week-old male and female expressing enhanced green fluorescence protein (EGFP) under the control of the forkhead box P3 (*Foxp3*) promoter (*Foxp3^EGFP* mice) were fed a purified Western diet deficient in vitamin A (WD-VAD) and injected with antisense oligonucleotide targeting the low-density lipoprotein receptor (ASO-LDLR) once a week for 16 wk to induce the development of atherosclerosis. After 16 wk, a group of mice was harvested (baseline) and the rest of the mice were injected once with sense oligonucleotide (SO-LDLR) to block ASO-LDLR (resolution). Mice undergoing resolution were either kept on the same diet (resolution–control) or switched to a Western diet supplemented with 50 mg/kg of β-carotene (resolution–β-carotene) for three more weeks. An additional group of mice fed with β-carotene was injected twice before sacrifice with anti-CD25 monoclonal antibody to deplete Treg (resolution–β-carotene+anti-CD25). The rest of the resolution groups were injected with IgG isotype control antibody. To quantify macrophage egress and monocyte recruitment, we injected a dose of EdU at week 15 and fluorescently labeled beads 2 d before harvesting the mice, respectively (see 'Methods' for details). (**A**) Representative images for macrophage (CD68⁺, top panels) and picrosirius staining to identify collagen using the bright-field (middle panels) or polarized light (bottom panels). Size bar = 200 µm. (**B**) Plaque size, (**C**) relative CD68 content, and (**D**) collagen content in the lesion. (**E**) Descriptive discriminant analysis employing the relative CD68 and collagen contents in the lesion as variables highlighting the false discovery rate (FDR) for each comparison. (**F**) Representative confocal image showing arginase 1 (green), CD68 (red), and DAPI (blue) in the lesion. (**G**) Relative arginase 1 area in the lesion. (**H**) Representative confocal image showing Cyp26b1 (green), CD68 (red), and DAPI (blue) in the lesion. (**I**) Relative Cyp26b1 area in the lesion. (**J**) EdU⁺ macrophages were identified by the colocalization of EdU (green) and DAPI (blue) in CD68⁺ (red) cells. (**K**) Number of EdU⁺ macrophages in the lesion. (**L**) Newly recruited monocytes were identified and (**M**) quantified by the presence of beads (green) on the lesion. Size bars = 50 µm. Each dot in the plot represents an individual mouse (n = 5–11 mice/group). Values are represented as means ± SEM. Statistical differences were evaluated

*Figure 5 continued on next page*

*Figure 5 continued*

using one-way ANOVA with Tukey's multiple comparisons test. Differences between groups were considered significant with a p-value<0.05. *p<0.05; **p<0.01; ***p<0.005; ****p<0.001.

The online version of this article includes the following figure supplement(s) for figure 5:

**Figure supplement 1.** Four-week-old male and female expressing enhanced green fluorescence protein (EGFP) under the control of the forkhead box P3 (*Foxp3*) promoter (*Foxp3*[EGFP] mice) were fed a purified Western diet deficient in vitamin A (WD-VAD) and injected with antisense oligonucleotide targeting the low-density lipoprotein receptor (ASO-LDLR) once a week for 16 wk to induce the development of atherosclerosis.

## Discussion

Seminal studies showed that β-carotene delays atherosclerosis progression in various experimental models by reducing plasma cholesterol levels (*Relevy et al., 2015*; *Harari et al., 2008*; *Shaish et al., 1995*), while greater levels of circulating β-carotene and retinoic acid are associated with lower risk of cardiovascular disease in people (*Wang et al., 2014*; *Huang et al., 2018*). In 2020, we demonstrated that these effects depend on BCO1 activity in mice, and that a genetic variant in the *BCO1* gene is associated with a reduction in plasma cholesterol levels in people (*Zhou et al., 2020*; *Amengual et al., 2020*). Our study shows that β-carotene promotes atherosclerosis resolution in two independent experimental models in a BCO1-dependent manner. Treg depletion studies revealed that dietary β-carotene favors Treg expansion in resolving lesions. Together, these findings identify dietary β-carotene and its conversion to vitamin A as a promising strategy to ameliorate plaque burden not only by delaying atherosclerosis progression, but also by reversing atherosclerosis inflammation.

Carotenoids are the main source of vitamin A in human diet, and the only source of this vitamin in strict vegetarians (*Grune et al., 2010*). Among those carotenoids with provitamin A activity, β-carotene is the most abundant in our diet and the only compound capable of producing two vitamin A molecules. Vitamin A is required for vision, embryo development, and immune cell maturation, making β-carotene a crucial nutrient for human health (*Coronel et al., 2019*). Unlike preformed vitamin A, the intestinal uptake of carotenoids is a protein-mediated process that depends on vitamin A status. This regulatory mechanism prevents the excessive uptake of provitamin A carotenoids to avoid vitamin A toxicity (*Lobo et al., 2013*; *Seino et al., 2008*).

BCO1 is the only enzyme in mammals capable of synthesizing vitamin A from carotenoids (*von Lintig and Vogt, 2000*; *Amengual et al., 2013*). Therefore, it is not surprising that SNPs in the *BCO1* gene are associated to alterations in circulating carotenoids including β-carotene (*Clifford et al., 2013*; *Yabuta et al., 2016*; *Moran et al., 2019*). We recently described that subjects harboring at least a copy of rs6564851-T allele, which increases BCO1 activity (*Lietz et al., 2012*), show a reduction in total cholesterol and non-HDL-C in comparison to those with two copies of the rs6564851-G variant (*Amengual et al., 2020*). A recent study revealed that this variant is also associated with triglyceride levels in middle-aged individuals (*León-Reyes et al., 2022*), implicating BCO1 activity as a novel regulator of plasma lipid profile. These studies provide a clinical relevance to studies performed in rodents and rabbits in which β-carotene-rich diets reduce plasma cholesterol and mitigate the development of atherosclerosis (*Zhou et al., 2020*; *Relevy et al., 2015*; *Harari et al., 2008*; *Shaish et al., 1995*).

Carotenoids, including β-carotene, possess antioxidant properties in lipid-rich environments (*Rodriguez-Concepcion et al., 2018*). The development of *Bco1*[-/-] mice of contributed to solving a long-lasting controversy in the carotenoid field by dissecting the biological effects of intact β-carotene and its role in vitamin A formation. When *Bco1*[-/-] mice are fed β-carotene, these mice accumulate large amounts of this compound in tissues and plasma (*Hessel et al., 2007*). In 2011, we reported that wild-type mice fed β-carotene exhibited a reduction in adipose tissue size. *Bco1*[-/-] mice subjected to the same experimental conditions did not show differences in adipose tissue size in comparison to control-fed mice despite accumulating large amounts of β-carotene in this tissue (*Coronel et al., 2022*). We recently overexpressed BCO1 in the adipose tissue of *Bco1*[-/-] mice fed β-carotene. Under these conditions, we observed an increase in vitamin A levels including retinoic acid and a reduction in adipose tissue size (*Coronel et al., 2022*). These effects are in line with those reported by Palou's group utilizing both dietary vitamin A and retinoic acid supplementation, which promotes fatty acid oxidation and thermogenesis in various experimental models (*Bonet et al., 2020*; *Amengual et al., 2018*; *Granados et al., 2013*; *Amengual et al., 2012*; *Amengual et al., 2010*; *Amengual et al.,*

*2008*; *Mercader et al., 2006*; *Felipe et al., 2005*; *Felipe et al., 2004*; *Puigserver et al., 1996*; *Berry and Noy, 2009*).

In 2020, we confirmed the implication of BCO1 activity and vitamin A in the development of atherosclerosis. We compared the effect of β-carotene on *Ldlr*$^{-/-}$ and *Ldlr*$^{-/-}$*Bco1*$^{-/-}$ mice using the same experimental diets utilized in this study (*Supplementary file 1*). In alignment with our clinical data, β-carotene reduced total cholesterol and non-HDL-C in *Ldlr*$^{-/-}$ mice, but not in *Ldlr*$^{-/-}$*Bco1*$^{-/-}$ mice (*Zhou et al., 2020*; *Amengual et al., 2020*). β-Carotene supplementation during atherosclerosis resolution, either in Western or standard diets, failed to improve plasma lipids in comparison to resolution–control groups (*Figures 1B and 2A*). This discrepancy could be attributed to the duration of the β-carotene feedings: 12 wk in our progression study vs. 3–4 wk in our resolution studies (*Zhou et al., 2020*). Another possibility is that the effect of β-carotene is only evident in mice displaying elevated plasma lipid levels such as reported in studies using *Ldlr*$^{-/-}$ and apolipoprotein E-deficient mice fed atherogenic diets (*Zhou et al., 2020*; *Relevy et al., 2015*; *Harari et al., 2008*).

In our atherosclerosis progression study, we attributed the effects of dietary β-carotene on plasma lipids to the formation of retinoic acid in the liver. We showed that retinoic acid treatment decreased cholesterol and triglyceride hepatic secretion rates in cultured hepatocytes and wild-type mice, suggesting that the effects of β-carotene on plasma lipid profile depend on the lipidation of hepatic lipoprotein particles (*Zhou et al., 2020*). Whether BCO1 activity is associated with the development of atherosclerotic cardiovascular disease in humans, however, remains unexplored.

Besides its effects on lipid metabolism, retinoic acid modulates immune cell function (*Miller et al., 2020*). Exogenous retinoic acid promotes alternative macrophage activation in various cell culture models, and skews naïve T cells to anti-inflammatory Tregs by directly upregulating the transcription factor FoxP3 (*He et al., 2021*; *Mucida et al., 2007*). The expression of FoxP3 is necessary and sufficient for the anti-inflammatory phenotype of Tregs (*Fontenot et al., 2005*; *Hori et al., 2003*). Studies carried out by Loke's group highlighted the interplay between alternative macrophage activation and Treg differentiation, a process that was proposed to be mediated by the production and release of retinoic acid by the macrophage (*Gundra et al., 2014*; *Broadhurst et al., 2012*). This hypothesis has not been demonstrated to date, partially due to the technical difficulty to quantify retinoic acid production in biological samples (*Kane et al., 2008*). We recently overcame this limitation and demonstrated for the first time that alternatively activated macrophages produce and release retinoic acid in a STAT6-dependent manner (*Pinos et al., 2023*). We also demonstrated that exogenous retinoic synergizes with interleukin 4 (IL4) to stimulate the expression of arginase 1, which is considered an anti-inflammatory marker in murine macrophages (*Pinos et al., 2023*; *Murray et al., 2014*). Arginase 1 catalyzes the production of ornithine, which is a precursor of proline (*Szondi et al., 2021*). Together with its derivative hydroxyproline, proline is a key limiting amino acid in the synthesis of collagen. These results indicate that the combination of retinoic acid with anti-inflammatory signals typically present during regressing lesions could favor atherosclerosis resolution by stimulating collagen deposition by the macrophage.

Our PCR data in liver homogenates show β-carotene supplementation upregulates Cyp26a1 expression in the liver, which, together with Cyp26b1 and Cyp26c1, are widely considered as surrogate markers for retinoic signaling in tissues (*Figure 1—figure supplement 1B*, *Abu-Abed et al., 2002*; *Ross and Zolfaghari, 2011*). Cyp26 family members are responsible for the degradation of intracellular retinoic acid, and their expression is regulated by retinoic acid response elements in their promoter (*Abu-Abed et al., 2002*; *Ross and Zolfaghari, 2011*). To examine whether β-carotene also stimulates retinoic acid signaling in the atherosclerotic lesion, we stained the lesions of *Foxp3*$^{EGFP}$ mice with Cyp26b1 (*Figure 5H and I*). Among the three Cyp26s, we selected Cyp26b1 based on our recent RNAseq data in murine bone marrow-derived macrophages exposed to retinoic acid (*Pinos et al., 2023*). We observed that Cyp26b1 was the only Cyp26 family member responsive to retinoic acid in macrophages. Our immunostaining showed that Cyp26b1 co-localizes with the macrophage marker CD68, in agreement with results obtained in human lesions showing that CYP26B1 are present in macrophage-rich areas (*Krivospitskaya et al., 2012*). This study also reported an association between aggravated atherosclerosis with an SNP in the *CYP26B1* gene with a greater retinoic acid catabolic activity (*Krivospitskaya et al., 2012*). These data suggest that greater retinoic acid levels in the lesion could decrease atherogenesis in humans, in agreement with our preclinical findings.

Despite the upregulation of Cyp26b1 expression, Treg expansion in regressing lesions in response to β-carotene was partially suppressed in mice injected with anti-CD25 (*Figure 4H*). To rule out intact β-carotene as the driving factor responsible for these effects, we quantified the number of Tregs in our experiment using *Bco1*[-/-] mice. As expected, both resolution groups displayed a greater number of Tregs in comparison to baseline mice, although we did not observe any effect in response to β-carotene (*Figure 3—figure supplement 1D and E*). These data suggest that vitamin A formation is required for Treg expansion in the atherosclerotic lesion of mice. To our knowledge, this is the first report showing that a dietary intervention with a nutrient administered at physiological concentrations can modulate Treg levels in the atherosclerotic lesion. As a reference, our diets were supplemented with 50 mg of β-carotene/kg in comparison to 80 mg of β-carotene/kg in carrots (USDA Database).

Our HPLC analyses in liver and plasma revealed that our experimental approach did not result in vitamin A deficiency despite feeding our mice vitamin A-deficient diets for several weeks to develop atherosclerosis. This was not surprising since mice are notably resistant to vitamin A deficiency. Protocols to develop vitamin A deficiency in wild-type mice require feeding the dams a vitamin A-deficient diet during pregnancy through weaning (*Gundra et al., 2017*), although the use of genetic models lacking key proteins in vitamin A transport, uptake, or storage are more adequate and time-efficient (*Quadro et al., 1999*; *O'Byrne et al., 2005*; *Amengual et al., 2014*). Future experiments utilizing these genetic models could shed light on the role of vitamin A deficiency on Treg expansion and its consequence on atherosclerosis resolution.

To establish a direct link between the effect of β-carotene supplementation and Tregs in our experimental model, we decided to deplete Tregs in *Foxp3*[EGFP] mice by infusing them with anti-CD25, following Sharma and colleagues' approach (*Sharma et al., 2020*). However, our data show that this strategy failed to completely deplete Tregs as previously reported by other investigators (*Couper et al., 2007*; *Hayes et al., 2020*). CD25[-] Tregs remained in circulation and tissues including aortic lesions (*Figure 4E–H*, *Figure 4—figure supplement 1*). It is possible that the conversion of β-carotene to retinoic acid favored Treg expansion by specifically favoring CD25[-] Treg proliferation, which are insensitive to anti-CD25.

CD25 is a crucial component of the IL2 receptor and it is expressed in CD4[+] T cell subsets other than Tregs (CD4[+]CD25[+]FoxP3[-]) (*Wing et al., 2002*). Therefore, strategies targeting FoxP3[+] cells directly are an attractive approach to specifically dissect the role of Tregs in our experimental model. For example, the use of *Foxp3*[DTR] mice (#016958, Jackson Labs), which express the diphtheria toxin receptor, would allow us to specifically deplete FoxP3[+] cells independently of the expression of CD25 (*Kim et al., 2007*). This approach would also eliminate CD8[+]CD25[+]FoxP3[+] Tregs, a relatively new Treg subpopulation (*Churlaud et al., 2015*). However, Treg depletion in *Foxp3*[DTR] mice requires injections with diphtheria toxin, which can result in potential adverse effects under certain experimental conditions (*Christiaansen et al., 2014*). Future studies dissecting the contribution of T cell subpopulations and the role vitamin A and other nutrients could play in their differentiation are needed to fully understand their role during atherosclerosis resolution.

Whether β-carotene supplementation during atherosclerosis resolution alters Treg content in other organs and disease models was outside the scope of this study. For example, an increase in Treg number in developing tumors could result in the reprogramming of pro-inflammatory macrophages toward resolving macrophages that could contribute to tumor expansion (*Ohue and Nishikawa, 2019*). In summary, partial Treg depletion was sufficient to mitigate the effects of β-carotene on plaque composition during the resolution of atherosclerosis, providing a direct link between β-carotene and Treg number (*Figure 5A–I*).

Another question remains unanswered: what is the source of vitamin A that favors Treg expansion in the lesion? Our HPLC analyses show that circulating β-carotene in wild-type mice is marginal, although hepatic vitamin A stores highlight an increase in vitamin A formation (*Figures 1H and 3A and B*). It is not clear whether naïve T cells express BCO1, which would enable them to locally produce vitamin A and its derivative retinoic acid. Previous work has related tissue macrophages as responsible for retinoic acid production and suggested that these cells would release it to signal nearby naïve T cells, which, in turn, could differentiate into Tregs. Retinoic acid could have originated from β-carotene or retinol/retinyl esters stored in the cell, although our RNA-sequencing analysis revealed that BCO1 expression in plaque macrophages is relatively limited (*Zhou et al., 2020*). Plaque macrophages, and T cells to a lesser extent, express various scavenger receptors that have been linked to the uptake of

retinoids and carotenoids such as the scavenger receptor class BI, lipoprotein lipase, and the cluster of differentiation 36 (*Reboul, 2019*). Determining whether naïve T cells rely on plaque macrophages to obtain retinoids and activate FoxP3 to differentiate into Treg is not clear to this day.

In summary, we report for the first time that vitamin A production from β-carotene favors Treg expansion in the atherosclerotic lesion, a process that contributes to atherosclerosis resolution. Together with the cholesterol-lowering effects of β-carotene described in the past, it unveils a dual role of dietary β-carotene and the enzyme BCO1: (1) BCO1 delays atherosclerosis progression by reducing plasma cholesterol and (2) favors atherosclerosis resolution by modulating Treg levels and macrophage polarization status.

## Methods

### Animal husbandry and diets

All procedures were approved by the Institutional Animal Care and Use Committees of the University of Illinois at Urbana Champaign. For all our studies, we utilized comparable number of male and female wild-type, *Ldlr*-/- (#002207, Jackson Labs, Bar Harbor, ME), *Bco1*-/- (*Hessel et al., 2007*), and *Foxp3*EGFP mice (#006772, Jackson Labs). All mice were in C57BL/6J background. Mice were kept under controlled temperature and humidity conditions with a 12 hr light/dark cycle and free access to food and water. Mice were weaned at 3 wk of age onto a breeder diet (Teklad global 18% protein diet: Envigo, Indianapolis, IN) in groups of 3–4 mice.

Control and β-carotene diets contained either placebo beadlets or β-carotene beadlets at a final concentration of 50 mg β-carotene/kg diet. For reference, the content of β-carotene in carrots is 80 mg/kg (USDA Database). Beadlets were a generous gift from DSM Nutritional Products (Sisseln, Switzerland). All diets were prepared by Research Diets (New Brunswick, NJ) by cold extrusion. The exact composition of all the diets used for this study is provided in *Supplementary file 1*. For all the experiments, we stimulated the development of atherosclerosis with a Western diet deficient in vitamin A (WD-VAD), as done in the past (*Zhou et al., 2020*; *Amengual et al., 2020*).

### Blood sampling and tissue collection

Before tissue harvesting, mice were deeply anesthetized by intraperitoneal injection with a mixture of ketamine and xylazine at 80 and 8 mg/kg body weight, respectively. We collected blood by cardiac puncture using ethylenediaminetetraacetic acid (EDTA)-coated syringes. We then perfused the mice with 10% sucrose in 0.9% sodium chloride (NaCl)-saline solution prior to tissue harvesting. Tissues were immediately snap-frozen in liquid nitrogen and kept at –80°C.

Aortic roots were collected after removing fat under a binocular microscope, embedded in optimum cutting temperature compound (OCT, Tissue-Tek, Sakura, Torrance, CA) and kept at –80°C.

### Antisense oligonucleotide targeting LDLR expression (ASO-LDLR)

To transiently deplete LDLR expression in wild-type, *Bco1*-/-, and *Foxp3*EGFP mice, we administered weekly an intraperitoneal injection containing 5 mg/kg body weight of ASO-LDLR for a period of 16 wk. At the end of the treatment, we injected a single dose of 20 mg/kg of the sense oligonucleotide (SO-LDLR) antidote, which binds and deactivates the ASO-LDLR (*Basu et al., 2018*). All oligonucleotide treatments were generously provided by Ionis Pharmaceuticals (Carlsbad, CA).

### HPLC analyses of carotenoids and retinoids

Nonpolar compounds were extracted from 100 μl of plasma or 30 mg of liver under a dim yellow safety light using methanol, acetone, and hexanes. The extracted organic layers were then pooled and dried in a SpeedVac (Eppendorf, Hamburg, Germany). We performed the HPLC with a normal phase Zobax Sil (5 μm, 4.6 × 150 mm) column (Agilent, Santa Clara, CA). Isocratic chromatographic separation was achieved with 10% ethyl acetate/hexane at a flow rate of 1.4 ml/min. For molar quantification of β-carotene and retinoids, we scaled the HPLC with a standard curve using the parent compound.

## Plasma lipid analyses

We measured plasma total cholesterol, cholesterol in the high-density lipoprotein fraction (HDL-C), and triglyceride levels using commercially available kits (FUJIFILM Wako Diagnostic, Mountain View, CA), according to the manufacturer's instructions.

Plasma lipoproteins were fractionated using FPLC with two Superose 6 10/300 GL columns (G.E. Healthcare, Boston, MA) on a Shimadzu HPLC system (Columbia, MD), as done in the past (*Zhou et al., 2020*; *Amengual et al., 2021*). Pooled samples (five mice/group) were used for analysis, and lipoprotein fractions were identified by quantifying cholesterol and triglyceride levels in each fraction.

## RNA isolation and RT-PCR

We isolated the total RNA using TRIzol reagent (Thermo Fisher Scientific, Waltham, MA) and Direct-zol RNA Miniprep kit (Zymo Research, Irvine, CA) according to the manufacturer's instructions. RNA was reverse-transcribed to cDNA using high-capacity cDNA reverse transcription kit (Applied Biosystems, Foster City, CA). Then we used TaqMan Master Mix (Applied Biosystems) to perform RT-PCR using the StepOnePlus RT-PCR system (ABI 7700, Applied Biosystems). We calculated the relative gene expression using the ΔCt method and normalized the data to β-actin (*Actb*). Probes (Applied Biosystems) include mouse *Ldlr* (Mm00440169_m1), mouse cytochrome P450 26a1 (*Cyp26a1*, Mm00514486_m1), and mouse *Actb* (Mm02619580_g1).

## Western blot analysis

For the determination of hepatic levels of LDLR, proteins were extracted from liver lysates in RIPA buffer (50 mM Tris pH 7.4, 150 mM NaCl, 0.25% sodium deoxycholate, 1% Nonidet P-40) in the presence of protease inhibitors. Total protein amounts were quantified using the Pierce BCA Protein Assay Kit (Thermo Fisher Scientific). A total of 80 µg of protein homogenates were separated by SDS-PAGE and transferred onto PVDF membranes (Bio-Rad, Hercules, CA). Membranes were blocked with fat-free milk powder (5% w/v) dissolved in Tris-buffered saline (15 mM NaCl and 10 mM Tris/HCl, pH 7.5) containing 0.01% Tween 100 (TBS-T), washed, and incubated overnight at 4°C with mouse anti-LDLR (Santa Cruz Biotechnologies, Dallas, TX) and mouse anti-GAPDH (Thermo Fisher Scientific) as a housekeeping control. Infrared fluorescent-labeled secondary antibodies were prepared at 1:15,000 dilution in TBS-T with 5% fat-free milk powder and incubated for 1 hr at room temperature.

## Treg depletion in *Foxp3^EGFP* mice

To deplete Tregs, mice were injected twice with 250 µg of either the isotype control IgG (Ultra-LEAFPurified Rat IgG1, $\lambda$ Isotype Ctrl antibody no. 401916, BioLegend, CA) or PC61 anti-CD25 monoclonal antibody (Ultra-LEAF Purified anti-mouse CD25 antibody no. 102040, BioLegend, CA). The first treatment took place a day after the administration of SO-LDLR, and the second injection 2 wk after, following established protocols (*Sharma et al., 2020*).

## Monocyte/macrophage trafficking studies

One week before harvesting the baseline group, we injected the mice intraperitoneally with 4 mg/ml of 5-ethynyl-2'-deoxyuridine (EdU) (Invitrogen, Waltham, MA) with the goal of labeling circulating monocytes to assess macrophage retention among groups. To compare monocyte recruitment among groups, we labeled circulating monocytes by injecting the mice retro-orbitally with 1 µm diameter Flouresbrite flash red plain microspheres beads (Polysciences Inc, Warrington, PA) diluted in sterile PBS 24 hr before harvesting, regardless of the experimental group. We assessed the efficiency of EdU and bead labeling by flow cytometry 48 hr and 24 hr after injection by tail bleeding, respectively (*Weinstock and Fisher, 2019*).

## Flow cytometry

To assess the number of monocytes labeled with either EdU or fluorescent beads, we incubated blood samples with red blood cell lysis buffer (Thermo Fisher Scientific) and blocked unspecific bindings using anti-mouse CD16/CD32 (Mouse BD Fc Block, BD Biosciences, Franklin Lakes, NJ). Cells were then stained with FITC-conjugated anti-mouse CD45 (BioLegend), PE-conjugated anti-mouse CD115 (BioLegend), and PerCP-conjugated anti-mouse Ly6C/G (BioLegend, San Diego, CA) for 30 min on ice. Cells were fixed, permeabilized, and stained for EdU using Click-iT EdU Pacific Blue Flow Cytometry

Assay Kit (Invitrogen) following the manufacturer's instructions. Beads were detected with the 640 nm red laser using a 660/20 bandpass filter. Monocyte recruitment and egress were estimated based on monocyte labeling efficiency, following established protocols (*Basu et al., 2018*).

For Treg quantifications, spleens were mashed and filtered through a 100 μm sterile cell strainer (Thermo Fisher Scientific). Blood and spleen homogenates were incubated with red blood cell lysis buffer (Thermo Fisher Scientific) and subsequently blocked with anti-mouse CD16/CD32 (BD Biosciences). We stained the cells with Fixable Viability Dye eFluor 780 (eBioscience, San Diego, CA) and eFluor 506-conjugated anti-mouse CD45 (BioLegend), PE-conjugated anti-mouse CD3e (BioLegend), FITC-conjugated anti-mouse CD4 (BioLegend), and BV421-conjugated anti-mouse CD25 (BioLegend) for 30 min on ice. We washed the cells twice before fixing and permeabilizing cell membranes using the eBioscience FoxP3/Transcription Factor Staining Buffer Set (Invitrogen) for 1 hr at room temperature, followed by incubation with Alexa Fluor 647-conjugated anti-mouse FoxP3 antibody (BioLegend) for 30 min at room temperature. All samples were measured on a BD LSR II analyzer (BD Biosciences), and results were analyzed with the FCS express 5 software (De Novo Software, Pasadena, CA).

## Atherosclerotic lesion analysis

Six-micrometer-thick sections were fixed and permeabilized with ice-cold acetone, blocked, and stained with rat anti-mouse CD68 primary antibody (clone FA-11; Bio-Rad) followed by biotinylated rabbit anti-rat IgG secondary antibody (Vector Laboratories, Burlingame, CA). CD68+ area was visualized using a Vectastain ABC kit (Vector Laboratories). Sections were then counterstained with hematoxylin, dehydrated in an ethanol gradient, xylene, and mounted with Permount medium (Thermo Fisher Scientific). Images were acquired using Axioskop 40 microscope (Carl Zeiss, Jena, Germany). For collagen content, frozen sections were fixed and stained using picrosirius red (Polysciences). Sections were scanned using the Axioscan.Z1 microscope (Carl Zeiss) using both bright field and polarized light. Total lesion area, CD68+, and collagen-positive areas were quantified using ImageJ software (NIH).

Aortic root sections were fixed in 10% formalin, permeabilized with 1% Triton X-100 in phosphate buffer, and stained using anti-FoxP3 (clone MF-14; BioLegend), anti-GFP (Cat# ab290; Abcam), anti-CD25 (clone IL2R.1; Thermo Fisher Scientific), anti-arginase 1 (Cat# 16001-1-AP; Thermo Fisher Scientific), anti-Cyp26b1b (Cat# 21555-1-AP; Thermo Fisher Scientific), and anti-Ki67 (clone SP6; Abcam). EdU+ cells in the lesion were visualized using Click-iT EdU Imaging Kit (Invitrogen). Secondary antibodies conjugated to Alexa fluorophores were purchased from Molecular Probes (Life Technologies). Images were taken using a Zeiss Axioscan.Z1 slide scanner (Carl Zeiss). Quantifications were performed using the ImageJ software (NIH).

## Statistical analysis

Data represented as bar charts are expressed as mean ± standard error of the mean (SEM). Data were analyzed using GraphPad Prism software (GraphPad Software Inc, San Diego, CA) by one-way ANOVA, followed by Tukey's multiple comparisons test. Differences between groups were considered significant with an adjusted $p$-value<0.05. Sample sizes were estimated based on previous experiments performed in our lab using comparable experimental approaches (*Zhou et al., 2020*; *Amengual et al., 2021*; *Yu et al., 2017*). Experimental samples were codified to prevent bias during analyses. The relationship between the two dependent factors, CD68 and collagen contents, and group differences was evaluated by descriptive discriminant analysis. Briefly, models were fitted under null and alternative hypotheses and goodness of fit was evaluated using the likelihood ratio test with randomized inference (bootstrap; n = 1000 simulations). Simultaneous hypothesis testing was corrected using the Benjamini–Hochberg procedure (*Benjamini et al., 2001*). The resulting false discovery rate (FDR) was considered significant with a cut-off <0.05.

## Acknowledgements

This work was supported by the National Institutes of Health (R01HL147252 to JA) and the United States Department of Agriculture (W4002 to JA). Asma'a Albakri's PhD studies at UIUC were funded by The University of Jordan (Amman, Jordan).

## Additional information

### Funding

| Funder | Grant reference number | Author |
|---|---|---|
| National Heart, Lung, and Blood Institute | R01HL147252 | Jaume Amengual |
| National Institute of Food and Agriculture | w4002 | Jaume Amengual |
| The University of Jordan | | Asma'a Albakri |

The funders had no role in study design, data collection and interpretation, or the decision to submit the work for publication.

### Author contributions

Ivan Pinos, Johana Coronel, Data curation, Formal analysis, Investigation, Methodology, Project administration; Asma'a Albakri, Patrick McQueen, Data curation, Formal analysis, Investigation, Methodology, Writing – original draft; Amparo Blanco, Data curation, Formal analysis, Validation, Investigation, Visualization, Methodology, Project administration, Writing – review and editing; Donald Molina, JaeYoung Sim, Methodology; Edward A Fisher, Writing – review and editing; Jaume Amengual, Conceptualization, Resources, Data curation, Supervision, Funding acquisition, Validation, Investigation, Methodology, Writing – original draft, Project administration, Writing – review and editing

### Author ORCIDs

Edward A Fisher (iD) https://orcid.org/0000-0001-9802-143X
Jaume Amengual (iD) https://orcid.org/0000-0002-6077-3545

### Ethics

This study was performed in strict accordance with the recommendations in the Guide for the Care and Use of Laboratory Animals of the National Institutes of Health. All procedures were approved by the Institutional Animal Care and Use Committees of the University of Illinois at Urbana Champaign (Protocol number 21015).

Reviewer #1 (Public Review): https://doi.org/10.7554/eLife.87430.3.sa1
Reviewer #2 (Public Review): https://doi.org/10.7554/eLife.87430.3.sa2
Author response https://doi.org/10.7554/eLife.87430.3.sa3

## Additional files

### Supplementary files
- MDAR checklist
- Source data 1. Raw data for all figures.
- Supplementary file 1. Composition of the experimental diets utilized in the study.

### Data availability

All data generated or analyzed during this study are included in the manuscript and supporting files.

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
