## [Editor Report · eLife assessment]

This study presents an **important** conceptual advance of how vitamin A and its derivatives contribute to atherosclerosis. There is **solid** evidence for the contributions of specialized populations of T cells in atherosclerosis resolution, including use of multiple in vivo models to validate the functional effects. A limitation is the insufficient analysis of lesions, but the manuscript has been improved from the original preprint version and the overarching conclusions have been refined.

---

## [Referee Report · Reviewer #1 (Public Review)]

This is an interesting study by Pinos and colleagues that examines the effect of beta carotene on atherosclerosis regression. The authors have previously shown that beta carotene reduces atherosclerosis progress and hepatic lipid metabolism, and now they seek to extend these findings by feeding mice a diet with excess beta carotene in a model of atherosclerosis regression (LDLR antisense oligo plus Western diet followed by LDLR sense oligo and chow diet). They show some metrics of lesion regression are increased upon beta carotene feeding (collagen content) while others remain equal to normal chow diet (macrophage content and lesion size). These effects are lost when beta carotene oxidase (BCO) is deleted. The study adds to the existing literature that beta carotene protects from atherosclerosis in general, and adds new information regarding regulatory T-cells. However, the study does not present significant evidence about how beta-carotene is affecting T-cells in atherosclerosis. For the most part, the conclusions are supported by the data presented, and the work is completed in multiple models, supporting its robustness. However there are a few areas that require additional information or evidence to support their conclusions and/or to align with the previously published work.

Specific additional areas of focus for the authors:

The premise of the story is that b-carotene is converted into retinoic acid, which acts as a ligand of the ROR transcription factor in T-regs. The authors measure hepatic markers of retinoic acid signaling (retinyl esters, Cyp26a1 expression) but none of these are measured in the lesion, which calls into question the conclusion that Tregs in the lesion are responsible for the regression observed with b-carotene supplementation.

There does not appear to be a strong effect of Tregs on the b-carotene induced pro-regression phenotype presented in Figure 5. The only major CD25+ cell dependent b-carotene effect is on collagen content, which matches with the findings in Figure 1 +2. This mechanistically might be very interesting and novel, yet the authors do not investigate this further or add any additional detail regarding this observation. This would greatly strengthen the study and the novelty of the findings overall as it relates to b-carotene and atherosclerosis.

The title indicates that beta-carotene induces Treg 'expansion' in the lesion, but this is not measured in the study.

Revised manuscript:

In the revised manuscript, the authors provide quantification of an RA-responsive gene in the plaque as evidence that RA signalling is indeed elevated upon b-carotene supplementation. It is not reduced upon blocking CD25 (Tregs) which implies that other cells in addition to Tregs are impacted by b-carotene supplementation that favourably remodels the plaque. The authors properly account for this by tempering their conclusions and recognize that Tregs are only partially responsible for the plaque phenotype upon b-carotene supplementation.

The authors chose not to further investigate why b-carotene impacted collagen production, instead including a discussion point. In this reviewer's opinion, it is a missed opportunity but hopefully something that can be investigated further by others.

---

## [Referee Report · Reviewer #2 (Public Review)]

Pinos et al present five atherosclerosis studies in mice to investigate the impact of dietary supplementation with b-carotene on plaque remodeling during resolution. The authors use either LDLR-ko mice or WT mice injected with ASO-LDLR to establish diet-induced hyperlipidemia and promote atherogenesis during 16 weeks, and then they promote resolution by switching the mice for 3 weeks to a regular chow, either deficient or supplemented with b-carotene. Supplementation was successful, as measured by hepatic accumulation of retinyl esters. As expected, chow diet led to reduced hyperlipidemia, and plaque remodeling (both reduced CD68+ macs and increased collagen contents) without actual changes in plaque size. But, b-carotene supplementation resulted in further increased collagen contents and, importantly, a large increase in plaque regulatory T-cells (TREG). This accumulation of TREG is specific to the plaque, as it was not observed in blood or spleen. The authors propose that the anti-inflammatory properties of these TREG explain the atheroprotective effect of b-carotene, and found that treatment with anti-CD25 antibodies (to induce systemic depletion of TREG) prevents b-carotene-stimulated increase in plaque collagen and TREG.

An obvious strength is the use of two different mouse models of atherogenesis, as well as genetic and interventional approaches. The analyses of aortic root plaque size and contents are rigorous and included both male and female mice (although the data was not segregated by sex). Unfortunately, the authors did not provide data on lesions in en face preparations of the whole aorta.

Overall, the conclusion that dietary supplementation with b-carotene may be atheroprotective via induction of TREG is reasonably supported by the evidence presented. Other conclusions put forth by the authors (e.g., that vitamin A production favors TREG production or that BCO1 deficiency reduces plasma cholesterol), however, will need further experimental evidence to be substantiated.

The authors claim that b-carotene reduces blood cholesterol, but data shown herein show no differences in plasma lipids between mice fed b-carotene-deficient and -supplemented diets (Figs. 1B, 2A, and S3A). Also, the authors present no experimental data to support the idea that BCO1 activity favors plaque TREG expansion (e.g., no TREG data in Fig 3 using Bco1-ko mice).

As the authors show, the treatment with anti-CD25 resulted in only partial suppression of TREG levels. Because CD25 is also expressed in some subpopulation of effector T-cells, this could potentially cloud the interpretation of the results. Data in Fig 4H showing loss of b-carotene-stimulated increase in numbers of FoxP3+GFP+ cells in the plaque should be taken cautiously, as they come from a small number of mice. Perhaps an orthogonal approach using FoxP3-DTR mice could have produced a more robust loss of TREG and further confirmation that the loss of plaque remodeling is indeed due to loss of TREG.

---

## [Author Response]

The following is the authors’ response to the original reviews.

**eLife Assessment**
This study presents a valuable conceptual advance of how Vitamin A and its derivatives contribute to atherosclerosis. There is solid evidence invoking the contributions of specialized populations of T cells in atherosclerosis resolution, including use of multiple in vivo models to validate the functional effect. The significance of the study would be strengthened with more detailed interrogation of lesions composition and consolidation with previous work on the topic from human studies.

Answer: We thank the reviewers and editorial office for their comments and constructive criticism. Below we provide point by point responses to the comments and concerns, which include the issues of lesion composition and consolidation with human studies. We also proofread the manuscript and included information about the immunostaining procedures that were previously missing (Lines 199 – 206).

**Public Reviews**

**REVIEWER #1:**
This is an interesting study by Pinos and colleagues that examines the effect of beta carotene on atherosclerosis regression. The authors have previously shown that beta carotene reduces atherosclerosis progress and hepatic lipid metabolism, and now they seek to extend these findings by feeding mice a diet with excess beta carotene in a model of atherosclerosis regression (LDLR antisense oligo plus Western diet followed by LDLR sense oligo and chow diet). They show some metrics of lesion regression are increased upon beta carotene feeding (collagen content) while others remain equal to normal chow diet (macrophage content and lesion size). These effects are lost when beta carotene oxidase (BCO) is deleted. The study adds to the existing literature that beta carotene protects from atherosclerosis in general, and adds new information regarding regulatory T-cells. However, the study does not present significant evidence about how beta-carotene is affecting T-cells in atherosclerosis. For the most part, the conclusions are supported by the data presented, and the work is completed in multiple models, supporting its robustness. However there are a few areas that require additional information or evidence to support their conclusions and/or to align with the previously published work.Specific additional areas of focus for the authors:1. The premise of the story is that b-carotene is converted into retinoic acid, which acts as a ligand of the RAR transcription factor in T-regs. The authors measure hepatic markers of retinoic acid signaling (retinyl esters, Cyp26a1 expression) but none of these are measured in the lesion, which calls into question the conclusion that Tregs in the lesion are responsible for the regression observed with b-carotene supplementation.

Answer: We agree with the Reviewer’s comment, which prompted us to quantify the expression of the retinoic acid-sensitive maker Cyp26b1 in the atherosclerotic lesions. Cyp26b1, together with Cyp26a1 and c1, contain retinoic acid response elements (RAREs) in their promoter, and therefore, are highly sensitive to retinoic acid. Indeed, the mRNA/protein expression of Cyp26s are widely considered surrogate markers for retinoic acid levels in cells or tissues.

We typically use Cyp26a1 as a surrogate marker for retinoic acid signaling in the adipose tissue and the liver, as we did in this study. However, our RNA seq data in murine bone-marrow derived macrophages (mBMDMs) exposed to retinoic acid revealed that Cyp26b1 is the only Cyp26 family member responsive to retinoic acid (PMID: 36754230). Actually, Cyp26a1 or c1 were not expressed in our mBMDMs (data not shown). Unlike the M2 marker arginase 1, Cyp26b1 did not respond to IL-4 (Figure iA). Hence, Cyp26b1 is an adequate marker to evaluate retinoic acid signaling in the lesion of mice, rich in macrophages.

Before staining the lesions, we validated the Cyp26b1 antibody by staining mBMDMs exposed to retinoic acid (Figure iB).

**Author response image 1. sa3fig1:** (A) mBMDMs were divided in M0 or M2 (exposed to IL-4 for 24 h), and then treated with either DMSO or retinoic acid for 6 h before harvesting for RNA seq analysis. Exploring the RNA seq dataset, we identified Cyp26b1 as a RA-sensitive gene in mBMDMs (PMID: 36754230). (B) Validation of Cyp26b1 antibody in mBMDMs exposed to retinoic acid confirms the suitability of this antibody for measuring retinoic acid signaling in our experimental settings.

In the current version of the manuscript, we include the results of Cyp26b1 quantifications (Figure 5H, I), (Lines: 362 - 366). To put these findings in perspective to human studies, we discuss these results with the role human CYP26B1 plays in the atherosclerotic lesion (Lines: 450 - 464).

1. There does not appear to be a strong effect of Tregs on the b-carotene induced pro-regression phenotype presented in Figure 5. The only major CD25+ cell dependent b-carotene effect is on collagen content, which matches with the findings in Figure 1 +2. This mechanistically might be very interesting and novel, yet the authors do not investigate this further or add any additional detail regarding this observation. This would greatly strengthen the study and the novelty of the findings overall as it relates to b-carotene and atherosclerosis.

Answer: As the Reviewer points out, the effects of β-carotene on collagen content are more pronounced than those on CD68 content in the lesion. Indeed, we have observed the majority of the experiments in this manuscript.

Collagen accumulation in the lesion is a complex process, where smooth muscle cells secrete collagen and plaque macrophages (typically) degrade it. Matrix metalloproteases produced by macrophages contribute to the degradation of collagen, and studies show that retinoic acid regulates the expression of metalloproteinases in various cell types (PMID: 2324527, 24008270). We explored the expression of metalloproteases in macrophages exposed to retinoic acid in our mBMDM RNA seq, but we did not observe any significant result (data not shown).

Interestingly, M2 macrophages can secrete collagen by upregulating arginase 1 expression. In the current version of the manuscript, we acknowledge this in the results (Lines: 358-359) and in the discussion section (Lines: 443-449).

1. The title indicates that beta-carotene induces Treg 'expansion' in the lesion, but this is not measured in the study.

Answer: Following the suggestion by the Reviewer, we have re-worded the title to “β-carotene accelerates the resolution of atherosclerosis in mice”

**REVIEWER #2:**
Pinos et al present five atherosclerosis studies in mice to investigate the impact of dietary supplementation with b-carotene on plaque remodeling during resolution. The authors use either LDLR-ko mice or WT mice injected with ASO-LDLR to establish diet-induced hyperlipidemia and promote atherogenesis during 16 weeks, and then they promote resolution by switching the mice for 3 weeks to a regular chow, either deficient or supplemented with b-carotene. Supplementation was successful, as measured by hepatic accumulation of retinyl esters. As expected, chow diet led to reduced hyperlipidemia, and plaque remodeling (both reduced CD68+ macs and increased collagen contents) without actual changes in plaque size. But, b-carotene supplementation resulted in further increased collagen contents and, importantly, a large increase in plaque regulatory T-cells (TREG). This accumulation of TREG is specific to the plaque, as it was not observed in blood or spleen. The authors propose that the anti-inflammatory properties of these TREG explain the atheroprotective effect of b-carotene, and found that treatment with anti-CD25 antibodies (to induce systemic depletion of TREG) prevents b-carotene-stimulated increase in plaque collagen and TREG.1. An obvious strength is the use of two different mouse models of atherogenesis, as well as genetic and interventional approaches. The analyses of aortic root plaque size and contents are rigorous and included both male and female mice (although the data was not segregated by sex). Unfortunately, the authors did not provide data on lesions in en face preparations of the whole aorta.

Answer: We appreciate the positive comments on rigor. We considered displaying our data segregated by sex, although for some experiments, we did not have matching numbers of male and female mice, which could be distracting for the reader.The goal of our study was to analyze changes in plaque composition. Therefore, our experimental approach was designed to study atherosclerosis resolution (plaque composition changes, but not plaque size) instead of atherosclerosis regression (both plaque composition and size change). As expected, we did not observe differences in plaque size at the level of the atherosclerotic root for any of our experiments, which deterred us from quantifying plaque content by en-face in the aorta.

2.Overall, the conclusion that dietary supplementation with b-carotene may be atheroprotective via induction of TREG is reasonably supported by the evidence presented. Other conclusions put forth by the authors (e.g., that vitamin A production favors TREG production or that BCO1 deficiency reduces plasma cholesterol), however, will need further experimental evidence to be substantiated.

Answer: We apologize for the lack of clarity in the presentation of our results and overstating our conclusions. We have rephrased some of these conclusions in the results and discussion sections.

3.The authors claim that b-carotene reduces blood cholesterol, but data shown herein show no differences in plasma lipids between mice fed b-carotene-deficient and -supplemented diets (Figs. 1B, 2A, and S3A).

Answer: As Reviewer 2 points out, we did not observe changes in plasma cholesterol between mice undergoing Resolution in response to β-carotene. For clarity, we rephrased our plasma lipids results for each of our experimental designs (Lines: 230 – 236, 270 – 272, and 288-290). We also include a clarification in the discussion section about the differential effects of β-carotene on plasma lipids when mice undergo atherosclerosis progression and resolution. (Lines: 419 - 430).

1. Also, the authors present no experimental data to support the idea that BCO1 activity favors plaque TREG expansion (e.g., no TREG data in Fig 3 using Bco1-ko mice).

Answer: We appreciate the suggestion by the Reviewer 2. In the current version of the manuscript, we stained the aortic roots from Bco1-/- mice for FoxP3. We did not observe differences between Control and β-carotene resolution groups, in agreement with the results in plaque composition (CD68 and collagen contents). These new data strengthen our manuscript and now we included these results as a Supplementary Figure 3D, E. (Lines: 465 - 471).

5.As the authors show, the treatment with anti-CD25 resulted in only partial suppression of TREG levels. Because CD25 is also expressed in some subpopulation of effector T-cells, this could potentially cloud the interpretation of the results. Data in Fig 4H showing loss of b-carotene-stimulated increase in numbers of FoxP3+GFP+ cells in the plaque should be taken cautiously, as they come from a small number of mice. Perhaps an orthogonal approach using FoxP3-DTR mice could have produced a more robust loss of TREG and further confirmation that the loss of plaque remodeling is indeed due to loss of TREG.

Answer: We agree with the reviewer, and we rephrased the results and discussion to avoid overstating our findings. We now acknowledge a second experimental approach would help us confirm our findings employing a blocking antibody targeting CD25. We favored the use of anti-CD25 infusions over other depletion methods based on the experimental protocol carried out by our collaborators in which the examined the effect of Tregs on atherosclerosis regression (PMID: 32336197). The utilization of FoxP3-DTR mice would nicely complement our findings. In the current version of the manuscript, we discuss this alternative approach (Line : 491 - 501).

**Recommendations for the Authors**
All reviewers agreed that despite the claims of the title, there is no direct interrogation of Tregs or vitamin A signaling in lesions.The work does not consolidate well with the role of B-carotene in human heart disease. Additional discussion and synthesis are required to elaborate on the significance of the findings. For example, the idea of beta carotene supplementation for cardiovascular prevention has attracted attention for years but recent meta-analysis showed no benefit, and, if anything, an increase in cardiovascular events. The U.S. Preventive Services Task Force (USPSTF) went as far to recommend AGAINST the use of beta-carotene for the prevention of cardiovascular disease.In light of the above point and elife editorial policies, please revise the title to include species.

Answer: Thanks for your feedback. Carotenoid metabolism in mammals is complex, and establishing direct parallelisms between humans and rodents must be done with caution. For example, β-carotene supplementation in humans inevitably results in the accumulation of this compound in plasma, while in rodents, β-carotene is quickly metabolized to vitamin A. Our findings over the years reveal that the effects of β-carotene in mice derive exclusively from its role as vitamin A precursor.

In the current study, we confirm our previous work utilizing Bco1-/- mice, which are unable to produce vitamin A when fed β-carotene. Then, we observe that vitamin A promotes atherosclerosis resolution in mice independently of alterations in plasma cholesterol in two independent mouse models. Lastly, we utilized anti-CD25 blocking antibodies to deplete Tregs to establish a direct connection between dietary β-carotene/vitamin A and Tregs in the lesion. While this experimental approach failed to completely deplete Tregs, our morphometric assays indicates that these infusions were sufficient to partially mitigate the effect of β-carotene on atherosclerosis resolution.

Regardless, in the discussion section of our manuscript, we attempt to consolidate our preclinical studies with clinical data (Lines: 374 – 376, and 461 – 464).

We have also revised the title, as suggested by Reviewer 1. We also included “mice” in the title to align with the editorial policies of eLife.

**Reviewer #1:**
1.1. The authors need to measure retinoic acid signaling directly in the lesion and in Tregs to be able to draw the conclusion that b-carotene is directly activating Tregs to promote regression.

Answer: Please see comments above.

1.2. The authors to investigate the role of beta carotene on collagen production by T-regs.

Answer: Please see comments above.

**Reviewer #2 (Recommendations For The Authors):**
Major:2.1. If the authors still have frozen sections of the aortas from their Bco1-ko experiment, it should be trivial to look at plaque TREG contents to confirm that vitamin A production is indeed needed for the effect of b-carotene on plaque remodeling.

Answer: Please see comments above.

Minor:2.2. This reviewer wonders if the axis for lesion size in all figures is off by an order of magnitude. Most studies show aortic root lesions in the 10^5 um2 range, not in the 10^6 um2.

Answer: We apologize for this error. We have corrected the units in all our quantifications.

2.3. FPLC lipoprotein profiles would enhance the manuscript.

Answer: We have run FPLCs for the plasmas and included them in the results (Lines: 233 – 236). Data are presented in Figure 1C, D.

2.4.This reviewer could not cope with the thought that mice that are fed 16+ weeks a diet that is vitamin A-deficient did not become vit A-deficient (e.g., Fig. 1E). Perhaps the authors could elaborate a little on this in their discussion.

Answer: Mice are extremely resistant to vitamin A deficiency. A common protocol to achieve deficiency in mice requires feeding a vitamin A deficient diet to dams during their pregnancy and lactation to deplete new-born pups of vitamin A stores. Even in that situation, pups display enough vitamin A stores to sustain circulating vitamin A levels to those observed in wild-type mice. In the current version of the manuscript, we have included a paragraph in the discussion to cover this “interesting” aspect. (Lines: 476 – 483).